# An HD-ZIP I Transcription Factor DZHDZ32 Upregulates Diosgenin Biosynthesis in *Dioscorea zingiberensis*

**DOI:** 10.3390/ijms26094185

**Published:** 2025-04-28

**Authors:** Huan Yang, Yi Li, Zixuan Hu, Jiaru Li

**Affiliations:** State Key Laboratory of Hybrid Rice, College Life Sciences, Wuhan University, Wuhan 430072, China; yanghuan@whu.edu.cn (H.Y.); ly40005@whu.edu.cn (Y.L.); huzixuan@whu.edu.cn (Z.H.)

**Keywords:** *Dioscorea zingiberensis*, diosgenin, DZHDZ32, HD-ZIP I, regulation

## Abstract

Diosgenin, a crucial precursor for steroidal drug production, has poorly understood regulatory pathways. Diosgenin is the primary active component of *Dioscorea zingiberensis*. Notably, *D. zingiberensis* also possesses the highest diosgenin content among *Dioscorea* species, reaching up to 16.15% of dry weight. This study identified DZHDZ32 as a potential regulator of diosgenin biosynthesis in *D. zingiberensis* through transient overexpression. To validate its function, we developed an optimized genetic transformation method for *D. zingiberensis* and generated two *DZHDZ32*-overexpressing lines. The DZHDZ32 transcription factor belongs to the HD-ZIP I subfamily and is localized to the nucleus. Notably, overexpression of *DZHDZ32* resulted in a significant increase in its transcript levels in leaves (264.59- and 666.93-fold), leading to elevated levels of diosgenin and its biosynthetic intermediates, including cholesterol and β-sitosterol. Specifically, diosgenin content increased by 41.68% and 68.07%, cholesterol by 10.29% and 16.03%, and β-sitosterol by 12.33% and 19.49% in leaves compared to wild-type plants. Yeast one-hybrid and dual-luciferase assays demonstrated that DZHDZ32 directly binds to the promoters of *ACAT* and *GPPS1*, consistent with the significant upregulation of *ACAT* and *GPPS1* expression (3.69- and 4.87-fold and 4.75- and 6.53-fold, respectively) in the overexpressing lines. This study established an optimized genetic transformation method for *D. zingiberensis* and identified DZHDZ32 as a key regulator of diosgenin biosynthesis. The discovery of DZHDZ32 has significant implications for enhancing diosgenin production and advancing steroidal drug development.

## 1. Introduction

*D. zingiberensis* is a species belonging to the genus *Dioscorea* (family *Dioscoreaceae*) [1,2,3]. Its rhizomes are particularly rich in diosgenin, with concentrations reaching up to 16.15% of dry weight [1,4,5]. This high diosgenin content makes *D. zingiberensis* not only a valuable medicinal resource but also a critical raw material for steroidal drug production with significant industrial relevance [6,7,8,9]. In addition to its industrial importance, this species also demonstrates notable medicinal properties [10]. Traditionally, it has been used in the treatment of rheumatism and arthritis and exhibits anti-inflammatory and analgesic effects [1,10,11,12]. Furthermore, it shows potential for managing cardiovascular diseases and regulating endocrine function [3,10,11]. Diosgenin is a type of steroidal sapogenin and an aglycone component of steroidal saponins [13]. Steroidal sapogenins are a class of compounds characterized by their steroid nucleus structure. Diosgenin, for instance, is a spirostane-type sapogenin. Diosgenin was first isolated from the tubers of *Dioscorea tokoro* [14], and in 1943, it was identified as a key precursor in the synthesis of steroidal hormone drugs [15]. Due to its cost-effective biosynthetic pathway, diosgenin remains an ideal starting material for steroid hormone production [16]. Despite its importance, the regulatory mechanisms underlying diosgenin biosynthesis remain largely unexplored. Some studies have shown that low concentrations of copper ions, ethylene, and jasmonic acid can enhance diosgenin accumulation [6,11,17,18,19]. The transcription factor DCWRKY11, which includes a WD (WRKY domain) in *Dioscorea composita* and is associated with the MeJA (methyl jasmonate) signaling pathway, has been reported to promote the expression of genes involved in steroidal saponin biosynthesis—the storage form of diosgenin [11]. Other transcription factors, including those from the WRKY, MYB (v-myb avian myeloblastosis viral oncogene homolog), bZIP (basic region-leucine zipper), bHLH (basic helix-loop-helix), NAC (NAM, ATAF, and CUC), HD-ZIP (homeodomain-leucine zipper), and AP2/ERF (APETALA2/ethylene responsive factor) families, may also play roles in diosgenin biosynthesis [8,11,20,21]. However, genetic transformation remains a significant challenge in *D. zingiberensis*, severely limiting molecular studies. Few studies have employed transgenic approaches in *D. zingiberensis* or closely related species, and investigations into transcriptional regulation remain scarce [22,23,24].

HD-ZIP transcription factors are widely distributed in plants and play vital roles in growth, development, and responses to environmental stressors [25,26,27,28]. However, their involvement in plant secondary metabolism, particularly in the biosynthesis of steroidal compounds, is not well understood. For example, in *Artemisia annua*, the HD-ZIP transcription factor AAHD1 regulates artemisinin accumulation [29]; in *Arabidopsis thaliana*, ATHB8 positively regulates brassinosteroid biosynthesis by modulating the expression of *CPD* (*constitutive photomorphogenic dwarf*), a key gene in the brassinosteroid pathway [30]. HD-ZIP transcription factors are also implicated in the regulation of steroidal alkaloid biosynthesis in *Veratrum mengtzeanum* [31]. HAT1, an HD-ZIP transcription factor in *A. thaliana*, can suppress the expression of the BR biosynthesis gene *BR6OX2* (*brassinosteroid-6-oxidase2*), thereby reducing BR content [32]. Currently, there are only speculative studies that lack experimental validation reporting that HD-ZIP transcription factors regulate the biosynthesis of diosgenin.

In this study, we use diosgenin as a model compound to investigate the regulatory role of HD-ZIP transcription factors. HD-ZIP proteins are divided into four subfamilies (I–IV) based on DNA-binding specificity [33,34], sequence similarity, conserved motifs, and physiological functions [35]. HD-ZIP I proteins are structurally the simplest, containing only the HD (homeodomain) and LZ (leucine zipper) domains located centrally within the protein [34,36,37], whereas HD-ZIP proteins from subfamilies II–IV possess more complex structures [33]. HD-ZIP I proteins typically bind to the CAAT(A/T)ATTG motif, though various sequence variants exist. The binding affinity is highest for the canonical CAAT(A/T)ATTG site [33,34,38] but may differ depending on sequence composition and complementarity at the ends [35]. HD-ZIP II and HD-ZIP III proteins share similar binding sites, but the central base in these sites might prefer (C/G). HD-ZIP IV proteins have distinctly different binding sites from the other subfamilies, with a specific binding site of TAAATG(C/T)A [34].

Although the diosgenin biosynthesis pathway in *D. zingiberensis* has been extensively studied [1], with considerable depth in understanding the synthesis mechanisms, current research suggests that HD-ZIP transcription factors may be involved in the biosynthesis of diosgenin [11,39]. Additionally, HD-ZIP transcription factors are capable of responding to various abiotic stresses and promoting the production of secondary metabolites [27,28,29,30,31,32,34,40,41,42]. However, the role of HD-ZIP transcription factors in *D. zingiberensis* has not been reported, and their function in diosgenin synthesis remains unclear. This study identifies a gene involved in diosgenin biosynthesis, clarifies its regulatory function, and elucidates the underlying mechanism, aiming to uncover the specific transcriptional regulation of diosgenin biosynthesis in *D. zingiberensis*. Understanding these regulatory elements will not only enhance diosgenin yield but also contribute to the broader understanding of secondary metabolite biosynthesis in plants.

## 2. Results

### 2.1. DZHDZ32 Promotes Diosgenin Accumulation in D. zingiberensis

Utilizing *D. zingiberensis* for a transient overexpression experiment, we identified the transcription factor DZHDZ32, a member of the HD-ZIP family in *D. zingiberensis*, which promotes diosgenin accumulation. The HD-ZIP family consists of 51 members in *D. zingiberensis*, named based on their amino acid lengths (Appendix A). We identified a critical transcription factor gene, *DZHDZ32*, a member of the HD-ZIP I subfamily in *D. zingiberensis* (Figure 1a,b). *Agrobacterium* carrying a pCXUN-4XHA empty vector was used to infiltrate WT (wild-type) leaves, resulting in an average diosgenin content of 5.44 mg/g. In contrast, leaves infiltrated with *Agrobacterium* harboring the *DZHDZ32*-containing pCXUN-4XHA vector had an average diosgenin content of 6.89 mg/g. This represents a 26.65% increase compared to the empty vector control (Figure 2a–c,e). qRT-PCR analysis confirmed that *DZHDZ32* transcript levels were upregulated by 42.55-fold in the transiently transformed leaves compared to the control (Figure 2d).

### 2.2. The Optimization of Genetic Transformation Methods for D. zingiberensis

This study established a more efficient and rapid genetic transformation system (Figure 3), with the advancement providing a more effective genetic transformation approach for the study of diosgenin biosynthesis and other secondary metabolite production in *Dioscorea* species. Genetic transformation within the *Dioscorea* genus has long been a challenge, with most functional gene studies relying on heterologous expression systems or unstable transformation methods [11,43]. Reports of stable genetic transformation using *Dioscorea* species as host plants are limited [23,24,25]. The main obstacles include low transformation efficiency and a prolonged period from *Agrobacterium* infection to the regeneration of complete transgenic plants (Table 1), which have hindered in-depth functional genomics research in this genus.

In our system, we achieved a transformation success rate of 46.8%, with transgenic plants regenerated in just four months (Figure 3). The buds of *D. zingiberensis* are infected with *Agrobacterium* carrying the *DZHDZ32* gene and then transferred to a co-cultivation medium for co-cultivation. Day 0 is the day when *Agrobacterium* infection (DAI) occurs. After 2 days of co-cultivation with the *Agrobacterium*, the bacterium is washed off, and the buds are transferred to a callus induction and selection medium to induce callus formation. By day 7 (7 DAI), noticeable tissue swelling can be observed, which becomes more pronounced by day 14 (14 DAI). At this point, the buds are transferred to a callus subculture and selection medium (which does not contain 2,4-D) to continue callus induction. Once the callus reaches the 35 DAI stage (Figure 3), it can be transferred to a regeneration and selection medium and placed under light. Approximately 4 weeks later, regenerated shoot points can be observed. When the regenerated shoots reach 91 DAI (Figure 3), they can be transferred to a rooting medium to induce rooting. About 2 weeks later (105 DAI), substantial regenerated roots can be observed, and after 4 weeks (119 DAI), the regenerated roots are fully developed. In the genus *Dioscorea*, transformation efficiencies vary greatly across different species. Specifically, the genetic transformation of *D. zingiberensis* is exceptionally difficult. Nevertheless, our research group has made significant breakthroughs in this area. Building on these advances, we have significantly reduced the time required while achieving higher transformation efficiency.

Apical and axillary buds of *D. zingiberensis* were used as explants due to their low contamination rate—only 1%—which is markedly lower than that observed in tissues such as seeds, flowers, or leaves. This low contamination rate provided a solid foundation for subsequent transformation experiments. During transformation, the addition of 2,4-D (2,4-dichlorophenoxyacetic acid) promoted dedifferentiation but inhibited redifferentiation of the tissues. In the early dedifferentiation stage, high concentrations of 2,4-D promoted callus formation from the buds. Once the callus was established, explants were transferred to a subculture medium containing NAA (naphthaleneacetic acid) and 6-BA (6-benzylaminopurine) but lacking 2,4-D to relieve the inhibitory effect and promote shoot regeneration. To accelerate the transformation process, buds were directly infected with *Agrobacterium* following sterilization, which reduced the time spent in the 2,4-D-containing callus induction medium. It is worth noting that the sterilization process can damage the buds. Using sterile seedlings as the explant source can lower bud mortality during sterilization, thereby further improving transformation efficiency. Compared to leaves, stems, and roots, the buds of *D. zingiberensis* not only exhibit lower contamination rates but also demonstrate faster and more efficient dedifferentiation, making them the optimal material for genetic transformation in this species.

*DZHDZ32* was cloned into the pCXUN-4XHA vector and introduced into *Agrobacterium tumefaciens* strain EHA105. Buds from WT plants were used for stable transformation. The infected buds underwent dedifferentiation and were subsequently redifferentiated to regenerate shoots and roots, ultimately yielding overexpressing transgenic plants (Figure 3).

Stable OE (*DZHDZ32*-overexpressing) lines were successfully generated. To further examine the relationship between *DZHDZ32* expression and diosgenin accumulation, we selected two transgenic lines with the highest diosgenin content and analyzed *DZHDZ32* expression in the roots, stems, leaves, and rhizomes. Expression analysis revealed the highest *DZHDZ32* transcript levels in the leaves, with varying levels of upregulation across other tissues (Figure 4a), indicating tissue-specific expression of *DZHDZ32* in *D. zingiberensis*.

Correspondingly, these transgenic lines exhibited significantly elevated diosgenin levels in all tissues compared to WT plants, with the most pronounced increase observed in rhizomes (Figure 4b). Specifically, diosgenin content in the rhizomes of the two transgenic lines increased by 68.41% and 89.59%, respectively, relative to WT. These findings support the hypothesis proposed by Li et al. 2022, that diosgenin biosynthesis primarily occurs in the leaves, while rhizomes serve as the main storage organs [1].

### 2.3. DZHDZ32 Promotes the Accumulation of Intermediate Metabolites in Diosgenin Synthesis

DZHDZ32 has been shown to enhance diosgenin accumulation. To further pinpoint the specific stages and regulatory nodes within the diosgenin biosynthetic pathway influenced by DZHDZ32, we analyzed two key intermediate metabolites: cholesterol and β-sitosterol, both of which are critical precursors in diosgenin synthesis [1,11,44,45]. Cholesterol levels were quantified via LC-MS in the roots, stems, leaves, and rhizomes of both OE lines and WT plants. The results revealed that *DZHDZ32* overexpression significantly increased cholesterol accumulation. Notably, the leaves—where diosgenin biosynthesis predominantly occurs—exhibited markedly higher cholesterol levels in the OE lines compared to WT. Moreover, cholesterol content was generally higher in leaf tissue than in other examined organs (Figure 4c). A similar trend was observed for β-sitosterol levels (Figure 4d). These findings suggest that DZHDZ32 may regulate diosgenin biosynthesis by acting upstream of sterol accumulation or possibly at multiple points that progressively enhance intermediate metabolite production throughout the biosynthetic pathway, ultimately promoting diosgenin synthesis.

### 2.4. Subcellular Localization of DZHDZ32

DZHDZ32 encodes a transcription factor that is typically localized in the nucleus [34,46]. To verify the subcellular localization of DZHDZ32, its amino acid sequence was analyzed using three different online prediction tools, all of which consistently predicted nuclear localization (Appendix A Appendix A). To experimentally confirm this prediction in *D. zingiberensis*, the *DZHDZ32* coding sequence was fused to EGFP (enhanced green fluorescent protein) and transiently expressed in young leaves. The empty vector PQG110-EGFP served as a control (Figure 5a).

Confocal microscopy analysis confirmed that DZHDZ32 is localized in the nucleus (Figure 5b–d). Notably, DZHDZ32 exhibited an early expression pattern, with strong nuclear fluorescence observed as early as 30 hpi (hours post-infiltration) (Figure 5b). Fluorescence intensity peaked at 33 hpi (Figure 5c), then gradually declined, with a noticeable reduction by 36 hpi (Figure 5d). By 48 hpi, the nuclear fluorescence signal of the DZHDZ32-EGFP fusion protein was markedly weaker compared to the control (Figure 5e).

### 2.5. Correlation Analysis Expression of DZHDZ32 with Genes Involved in the Diosgenin Biosynthetic Pathway

Given that diosgenin is predominantly synthesized in leaves and that *DZHDZ32* exhibits its highest expression in leaf tissues [1], along with the observation that diosgenin content is elevated to varying degrees in both *D. zingiberensis* overexpression lines, these findings suggest a potential involvement of *DZHDZ32* in the diosgenin biosynthetic pathway. To identify specific points of interaction, we analyzed the correlation between *DZHDZ32* expression and genes involved in diosgenin biosynthesis using transcriptome data (accession number CRA007729).

Under drought stress conditions, the expression of *DZHDZ32* exhibited strong positive correlations (Spearman correlation coefficient ≥ 0.8) with the expression levels of several key biosynthetic genes, including *ACAT* (*acetyl-CoA acetyltransferase gene*), *HMGR* (*3-hydroxy-3-methylglutaryl-CoA reductase gene*), *MVD* (*diphosphomevalonate decarboxylase gene*), *CMK* (*4-diphosphocytidyl-2-C-methyl-D-erythritol kinase gene*), *CAS* (*cycloartenol synthase gene*), *SMT1* (*sterol C-24 methyltransferase gene*), and cytochrome P450 genes *CYP94D143* and *CYP94A11*. Similarly, diosgenin content was strongly correlated with *GPPS1* (*geranylgeranyl pyrophosphate synthase gene*), *CAS*, *SMT2*, and *CYP51* (*sterol C-14 demethylase gene*) (Figure 6a).

Following MeJA treatment, the expression of *DZHDZ32* also displayed strong positive correlations with the expression levels of *ACAT*, *PMK* (*phosphomevalonate kinase gene*), *MVD*, *DXS* (*1-deoxy-D-xylulose-5-phosphate synthase 1 gene*), *CMK*, *SMT1*, *CYP72A12-4*, and *CYP94A11* (Figure 6b). Notably, the correlation coefficient between *DZHDZ32* expression and diosgenin content under MeJA treatment was 1.0, indicating a perfect positive correlation.

To determine whether these strongly correlated genes serve as functional links between *DZHDZ32* and the diosgenin biosynthetic pathway, we examined their expression levels in *DZHDZ32*OE lines compared with WT plants. As shown in Figure 6c, several genes were significantly upregulated or downregulated in the transgenic lines. Considering that only the upregulation of biosynthetic genes can promote diosgenin accumulation, we focused on those that were upregulated: *ACAT*, *MVD*, *GPPS1*, and *CYP72A12-4*.

Specifically, *ACAT* expression increased by 3.69- and 4.87-fold, *MVD* by 0.40- and 0.55-fold, *GPPS1* by 4.75- and 6.53-fold, and *CYP72A12-4* by 3.39- and 4.76-fold in the two *DZHDZ32*OE lines (Figure 6c). These genes likely represent key regulatory nodes through which *DZHDZ32* modulates the diosgenin biosynthetic pathway.

### 2.6. Promoter Analysis of Genes in the Diosgenin Biosynthetic Pathway Correlated with DZHDZ32

Analysis of the protein domain structure of DZHDZ32 revealed the presence of both an HD and an LZ domain, indicating that DZHDZ32 belongs to the HD-ZIP family. Phylogenetic tree analysis further classified DZHDZ32 within the HD-ZIP I subfamily, as it clustered with members of this subfamily from *A. thaliana* and *Oryza sativa* (Figure 1a,b).

HD-ZIP I transcription factors are known to bind specific DNA motifs, such as CAAT(A/T)ATTG in *Arabidopsis* [28,33]. Other reported binding sequences include CAATNATTG and ATTNAAT. In the tung tree, the HD-ZIP I family VFHB21 has been shown to bind to several motifs, including CAATAATTG, GAATGATTG, TAATTATTA, TAATAATTA, AAATTATTA, and TAATTATTG, with the highest binding affinity observed for TAATTATTA [38,47]. Given that DZHDZ32 promotes the expression of *ACAT*, *MVD*, *GPPS1*, and *CYP72A12-4* (Figure 7c), we analyzed the 2000 bp upstream promoter sequences of *ACAT* and *GPPS1* for putative DZHDZ32 binding sites. The results revealed multiple potential HD-ZIP I binding motifs within these promoter regions (Figure 7a), suggesting that DZHDZ32 may directly regulate the transcription of these genes.

### 2.7. DZHDZ32 Regulates Genes in the Diosgenin Biosynthetic Pathway

To determine whether DZHDZ32 directly interacts with the promoters of *ACAT* and *GPPS1*, Y1H assays were conducted. The results confirmed that DZHDZ32 could bind to the promoter regions of both genes (Figure 7a,b). To further validate this interaction, a dual-LUC assay was performed. Consistent with the Y1H results, DZHDZ32 significantly activated the promoter regions of *ACAT* and *GPPS1*, as evidenced by increased LUC/REN ratios (Figure 7c,d).

Taken together, these findings indicate that DZHDZ32 directly binds to the upstream regulatory regions of *ACAT* and *GPPS1*, thereby modulating their transcription and promoting diosgenin biosynthesis (Figure 8).

## 3. Discussion

Transcription factors play a crucial role in regulating diosgenin biosynthesis. Previous studies have suggested that members of the WRKY, MYB, bZIP, bHLH, NAC, HD-ZIP, and AP2/ERF families may be involved in this process [8,11,20,21]. In the present study, we identified DZHDZ32 as a key HD-ZIP transcription factor that regulates the biosynthesis and accumulation of diosgenin. Phylogenetic analysis classified DZHDZ32 as a member of the HD-ZIP I subfamily, the largest among the four HD-ZIP subfamilies.

HD-ZIP I transcription factors are known to respond to a wide range of abiotic stresses, including drought, salinity, cold, exposure to toxic metals/metalloids, and light [28,48,49,50,51,52,53,54]. They are also responsive to plant hormones such as ABA (abscisic acid), MeJA, and GA (gibberellin) [28,51,55,56,57]. In addition to their roles in stress signaling, HD-ZIP I factors influence metabolite accumulation, fruit abscission, and plant morphogenesis and structural development [34,38,53,58]. These diverse functions underscore the significance of HD-ZIP I transcription factors in plant growth, development, and adaptation.

Our study demonstrates the potential of DZHDZ32 to enhance diosgenin production, as evidenced by its elevated expression and strong correlation with diosgenin accumulation. This was further validated by the generation of *DZHDZ32* overexpression transgenic lines. Notably, genetic transformation in *D. zingiberensis* has historically been challenging due to the lack of a well-established transformation system, which has limited functional genomic studies in this species. The transformation method developed in this study significantly reduces the time required to obtain transgenic lines, enabling the generation of complete transgenic plantlets in approximately four months. Moreover, the improved transformation efficiency facilitates more efficient and comprehensive investigations into gene function in *D. zingiberensis*.

The *DZHDZ32*OE lines showed no obvious phenotypic differences compared to WT plants. However, diosgenin content increased by 41.68% and 68.07% in the leaves of the two transgenic lines, respectively. Given that diosgenin is a valuable secondary metabolite and a precursor for various steroidal drugs, this increase is of considerable significance. Concurrently, we observed upregulation of key biosynthetic genes, including *ACAT*, *MVD*, *GPPS1*, and *CYP72A12-4*, suggesting that DZHDZ32 positively regulates diosgenin synthesis through the modulation of these genes.

Subcellular localization experiments confirmed that DZHDZ32 is localized to the nucleus. Interestingly, DZHDZ32 protein expression was detectable as early as 33 hpi in *D. zingiberensis*, earlier than the typical 36–48 h observed for other proteins. Since diosgenin biosynthetic genes are nuclear-encoded, the early nuclear localization of DZHDZ32 supports its regulatory role in this pathway.

The binding specificity of HD-ZIP I transcription factors can vary across plant species, reflecting the structural diversity of these proteins. The most common binding motif is CAAT(A/T)ATTG, as observed in *A. thaliana* (e.g., ATHB1). However, other variants have been identified. For example, the HD-ZIP I factor VFHB21 from *Vernicia fordii* (tung tree) binds to multiple motifs, including CAATAATTG, GAATGATTG, TAATTATTA, TAATAATTA, AAATTATTA, and TAATTATTG [34]. A general consensus sequence for HD-ZIP I binding is considered to be NATTNATTN, with a central A or T nucleotide and complementary sequences at both ends enhancing binding affinity. These variations reflect the adaptability of HD-ZIP I proteins in recognizing diverse promoter elements and regulating a wide array of biological processes.

In our study, Y1H and dual-LUC assays confirmed that DZHDZ32 binds to the upstream promoter regions of diosgenin biosynthetic genes. Furthermore, *DZHDZ32* overexpression led to the upregulation of these genes in transgenic lines compared to WT plants. These results indicate that DZHDZ32 promotes diosgenin accumulation by directly binding to promoter regions of key biosynthetic genes, thereby enhancing their expression. This regulatory mechanism provides new insights into the transcriptional control of diosgenin biosynthesis in *D. zingiberensis*.

## 4. Materials and Methods

### 4.1. Plant Material Selection and Utilization

All experiments in this study used *D. zingiberensis* obtained from Li et al. 2022 [1] (WT, PRJNA716093). These plants were cultivated in a greenhouse at Wuhan University under long-day conditions (16 h light/8 h dark) at a constant temperature of 25 ± 1 °C. For transgenic experiments, the apical and axillary buds of WT were chosen as the experimental material. Both subcellular localization and dual-LUC experiments used young leaves of WT as injection materials, with three leaves of uniform size selected as biological replicates. Similarly, young leaves from *D. zingiberensis* were utilized as the source material for RNA extraction in qPCR (quantitative polymerase chain reaction) analyses.

### 4.2. Transgenic Overexpression of DZHDZ32 in D. zingiberensis

The pCXUN-4XHA vector containing *DZHDZ32* (pCXUN-*DZHDZ32*-4XHA) was constructed using a homologous recombination method and subsequently transferred into the *Agrobacterium* strain EHA105. The hygromycin antibiotic resistance gene present in pCXUN-4XHA was replaced with a kanamycin resistance gene through homologous recombination. Prior to infecting *D. zingiberensis*, the activated *Agrobacterium* was transferred into a suspension medium (Appendix A Appendix A) and incubated at 28 °C with shaking at 180 rpm for 3 to 3.5 h. The *Agrobacterium* suspension was then adjusted to a concentration corresponding to an OD600 value of 0.8. *Agrobacterium* adjusted to an optimal concentration was utilized for transient expression and genetic transformation. In this transient expression experiment, the suspension was incubated in the dark at 22 °C for 2 h. Following incubation, it was injected into the abaxial side of young leaves of *D. zingiberensis*. The leaves were then kept in the dark for 12–16 h, followed by exposure to light for 17–21 h before samples were collected.

For genetic transformation, the apical and axillary buds of the WT were surface sterilized by immersion in alcohol for 30 s, followed by disinfection in a 2% NaClO solution for 7 min. The buds were then rinsed three times with sterile water. The sterilized buds were immersed in the *Agrobacterium* suspension and subjected to infection for 30 min. Excess moisture on the surface of the buds was removed using sterile filter paper, and the buds were then placed under a layer of filter paper and dried on a super-clean workbench. The buds were transferred onto a co-cultivation medium overlaid with a layer of filter paper and cultured in the dark at 25 °C for two days on the co-cultivation medium (Appendix A Appendix A). Following this, the buds were washed seven times with sterile water and soaked in a solution containing 400 mg/L of timentin for 30 min. The buds were then transferred to a callus induction and selection medium (Appendix A Appendix A). After two weeks, the callus was subcultured onto a callus subculture and selection medium (Appendix A Appendix A). Following several weeks, the callus was transferred to a regeneration and selection medium (Appendix A Appendix A). After several weeks of growth on the regeneration and selection medium, the antibiotic-resistant shoots were transferred to a rooting medium (Appendix A Appendix A).

### 4.3. Determination and Analysis of Diosgenin and Its Intermediates Cholesterol and β-Sitosterol

Diosgenin was extracted from *D. zingiberensis* following previously described methods [1,11,59,60]. The roots, stems, leaves, and rhizomes from OE lines and WT plants were freeze-dried and ground into a fine powder. A 50 mg aliquot of each sample was subjected to acid hydrolysis with 2 mL of 1 M H_2_SO_4_ at 100 °C for 4 h. After centrifugation at 13,000 rpm for 10 min, the supernatant was discarded. The pellet was washed repeatedly with 1 M NaOH, followed by centrifugation at 13,000 rpm for 10 min each time until the pH of the supernatant reached approximately 7. The resulting precipitate was dried at 55 °C and reground into powder. One milliliter of isopropanol was added to the powder, followed by shaking at 200 rpm at 50 °C for 30 min. The mixture was then subjected to ultrasonication for 60 min and centrifuged at 13,000 rpm for 10 min. The supernatant was filtered through a 0.22 µm membrane and transferred to sample vials for analysis. Quantification was performed using UPLC-MS/MS (Thermo Scientific, Waltham, MA, USA), and data were processed using Xcalibur 4.1 software (Thermo Scientific, Waltham, MA, USA). Three biological replicates were analyzed for each sample.

Cholesterol and β-sitosterol were quantified as previously described [4]. Samples were lyophilized and ground into fine powder using a pulverizer. A 5 mg portion of the powder was extracted with 1 mL of a chloroform:methanol (2:1, *v*/*v*) mixture and incubated in a 75 °C water bath for 1 h. Subsequently, 250 μL of 6% methanolic KOH was added, and the mixture was incubated at 90 °C for 1 h, followed by 1 h of ultrasonication. The solution was filtered through a 0.22 μm membrane filter, and the filtrate was extracted three times with 500 μL of hexane. After each extraction, the samples were sonicated for 30 min and transferred to a new tube. The combined hexane extracts were evaporated to dryness under vacuum at room temperature using a vacuum concentrator. The residue was reconstituted in 500 μL of hexane for the quantification of cholesterol and β-sitosterol. Analysis was conducted using UPLC-MS/MS (Thermo Scientific, Waltham, MA, USA), with three biological replicates for each sample.

### 4.4. Identification and Phylogenetic Analysis of HD-ZIP Gene Family Members in D. zingiberensis

The genome of *D. zingiberensis* was retrieved from NCBI under the accession number PRJNA716093. HD-ZIP protein sequences from *A. thaliana* were downloaded from The *Arabidopsis* Information Resource (TAIR; https://www.arabidopsis.org/ (accessed on 18 October 2024)), and those from rice were obtained from the Rice Genome Annotation Project (https://rice.uga.edu/ (accessed on 20 October 2024)). To identify putative HD-ZIP proteins in *D. zingiberensis*, BLAST -2.16.0 searches were performed using HD-ZIP proteins from *Arabidopsis* and rice as queries, with an E-value cutoff of 1 × 10^−5^. Additionally, HMM (hidden Markov model) profiles for the conserved HD (PF00046) and LZ (PF02183) domains were obtained from the PFAM database (http://pfam.xfam.org/ (accessed on 21 October 2024)). The HMMER tool was used to screen the complete *D. zingiberensis* proteome for HD-ZIP proteins.

Candidate HD-ZIP proteins identified through both BLAST and HMMER searches were compiled, cross-referenced with genome annotation data, and further validated. Multiple sequence alignment of the confirmed HD-ZIP proteins was performed using ClustalW. A phylogenetic tree was constructed using the maximum likelihood method in MEGA7, with 1000 bootstrap replicates to assess evolutionary relationships.

### 4.5. Quantitative Real-Time PCR (qRT-PCR) for Diosgenin Biosynthesis Pathway Genes

Young leaves from WT plants were used for RNA extraction in qRT-PCR analysis. RNA isolation and cDNA synthesis were performed according to the methods described by Li et al. 2024 [4]. Total RNA was extracted using TRIzol reagent (TransGen, Beijing, China), and first-strand cDNA was synthesized using the HiScript III 1st Strand cDNA Synthesis Kit (Vazyme, Nanjing, China). qPCR amplification was conducted using PerfectStart Green qPCR SuperMix (TransGen, Beijing, China). Primers for key genes are listed in Appendix A. Relative gene expression was calculated using the 2^−ΔΔCT^ method, with *DZACTIN* as the internal reference gene. Each reaction was performed with three biological replicates.

### 4.6. Subcellular Localization

To investigate the subcellular localization of DZHDZ32, the coding sequence of *DZHDZ32* was cloned into the PQG110-EGFP vector, resulting in the construct PQG110-*DZHDZ32*-EGFP. This construct was introduced into *A. tumefaciens* strain EHA105. The transformed *Agrobacterium* was cultured in LB medium containing 50 mg/L kanamycin and 20 μM AS (acetosyringone) at 28 °C with shaking at 200 rpm for 8 h. The overnight culture was then resuspended in infiltration buffer containing 10 mM MES, 10 mM MgCl_2_, and 200 μM AS and adjusted to an OD600 of 0.8. The suspension was incubated in the dark at 22 °C for 2 h before being infiltrated into the abaxial side of young *D. zingiberensis* leaves.

Following infiltration, plants were kept in the dark for 12–16 h and then exposed to light for 17–21 h. The empty PQG110-EGFP vector was used as a negative control. Subcellular localization of the EGFP-tagged DZHDZ32 protein was examined 30–48 h post-infiltration using a laser scanning confocal microscope (Leica, Wetzlar, Hessen, Germany). DAPI staining was used to visualize the nucleus. All experiments were conducted with three biological replicates.

### 4.7. Correlation Analysis

Transcriptome data were obtained from the China National Center for Bioinformation (https://bigd.big.ac.cn/gsa (accessed on 28 October 2024)) under accession number CRA007729. The expression levels of *DZHDZ32* and genes involved in diosgenin biosynthesis were quantified using FPKM values. Expression data for diosgenin-related genes were sourced from a previous study [1]. Genes with an average FPKM value greater than 2 were selected for heat map generation. Correlation analysis was conducted and visualized using Origin 2024 software.

### 4.8. Yeast One-Hybrid (Y1H) Assay

The *DZHDZ32* gene was subcloned into the pGADT7 plasmid to generate the pGADT7-prey construct, while a 2000 bp upstream promoter sequence of the target gene was inserted into the pAbAi plasmid to produce the pAbAi-bait construct. These constructs were co-transformed into *Saccharomyces cerevisiae* strain Y187. The combination of p53-AbAi and pGADT7-*p53* served as a positive control, while p53-AbAi co-transformed with the empty pGADT7 vector was used as a negative control. Transformed yeast cells were plated on SD/-Leu/-Ura agar supplemented with 200 ng/mL Aureobasidin A (AbA). All assays were performed with three biological replicates.

### 4.9. Dual-Luciferase (Dual-LUC) Assay

Dual-luciferase assays were performed to assess whether *DZHDZ32* could activate the promoters of genes involved in diosgenin biosynthesis. The *DZHDZ32* coding sequence was cloned into the pGreenII62-SK vector as the effector, while the promoters of target genes were inserted upstream of the luciferase reporter gene in the pGreenII0800-LUC vector, serving as reporters. An empty pGreenII62-SK vector was used as a negative control. Effector and reporter constructs were individually transformed into *A. tumefaciens* strain EHA105. The transformed bacteria were cultured in LB medium containing 50 mg/L kanamycin and 20 μM AS at 28 °C with shaking at 200 rpm for 8 h. Cultures were then resuspended in infiltration buffer (10 mM MES, 10 mM MgCl_2_, and 200 μM AS) and adjusted to an OD600 of 0.8. The suspension was incubated in the dark at 22 °C for 2 h before being infiltrated into the abaxial side of young leaves. After infiltration, plants were kept in the dark for 12–16 h, then transferred to light for 17–21 h. LUC (luciferase) and REN (Renilla luciferase) activities were measured using a luciferase assay kit (Promega, Madison, WI, USA). All experiments were conducted with three biological replicates.

## 5. Conclusions

We have established an enhanced, rapid genetic transformation system for *D. zingiberensis*, which will significantly facilitate future studies on diosgenin biosynthesis and other valuable secondary metabolites in medicinal plants. A key discovery was that DZHDZ32 functions as a positive regulator of diosgenin production. Overexpression of *DZHDZ32* in transgenic lines led to elevated diosgenin levels, concomitant accumulation of pathway intermediates, and upregulation of biosynthetic genes. We further identified specific target genes of DZHDZ32 within the diosgenin pathway, demonstrating that DZHDZ32 directly enhances their expression.

Together, these results position DZHDZ32 as a pivotal regulator of diosgenin biosynthesis, offering a strategic target for increasing diosgenin yields and guiding the development of novel steroidal therapeutics. Moreover, this work provides a valuable framework for future investigations into the regulatory networks that control secondary metabolite production.

## Figures and Tables

**Figure 1 ijms-26-04185-f001:**
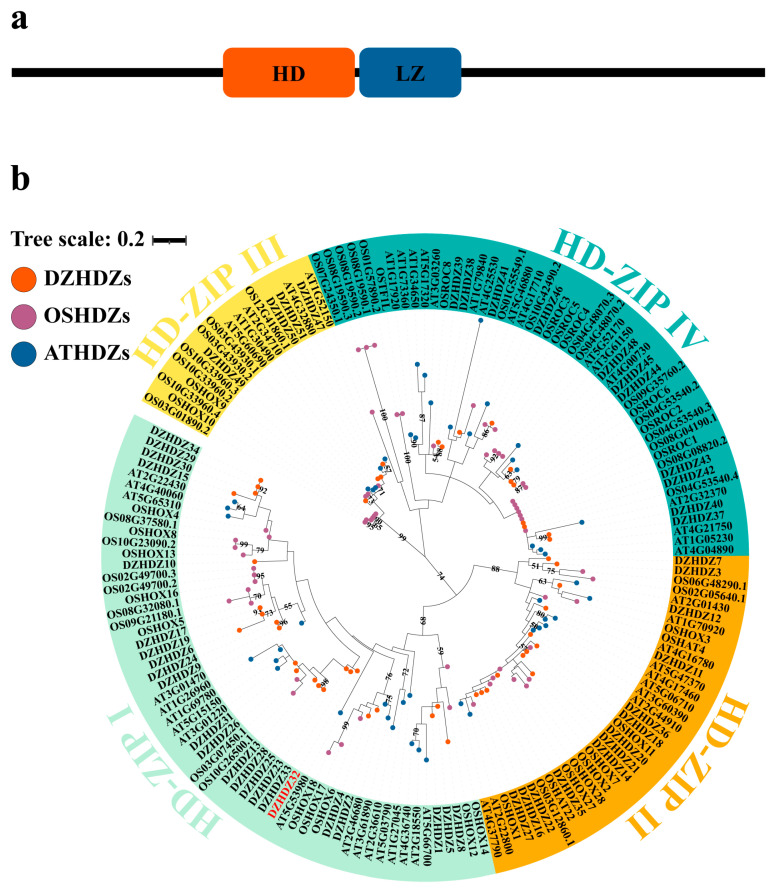
Protein structure of DZHDZ32 and phylogenetic analysis of the HD-ZIP family in *D. zingiberensis*. (**a**) Protein structure of DZHDZ32. The DZHDZ32 protein is composed of 302 amino acids and contains an HD and an LZ domain. The black line represents the DZHDZ32 protein, the orange rectangle represents the HD domain, and the blue rectangle represents the LZ domain DZHDZ32. (**b**) Phylogenetic analysis of the HD-ZIP family in *D. zingiberensis*. The red text indicates the key transcription factor DZHDZ32. Different color labels represent different subgroups. The HD-ZIP family of *D. zingiberensis* was clustered with HD-ZIP genes from *Arabidopsis* and rice into four subgroups.

**Figure 2 ijms-26-04185-f002:**
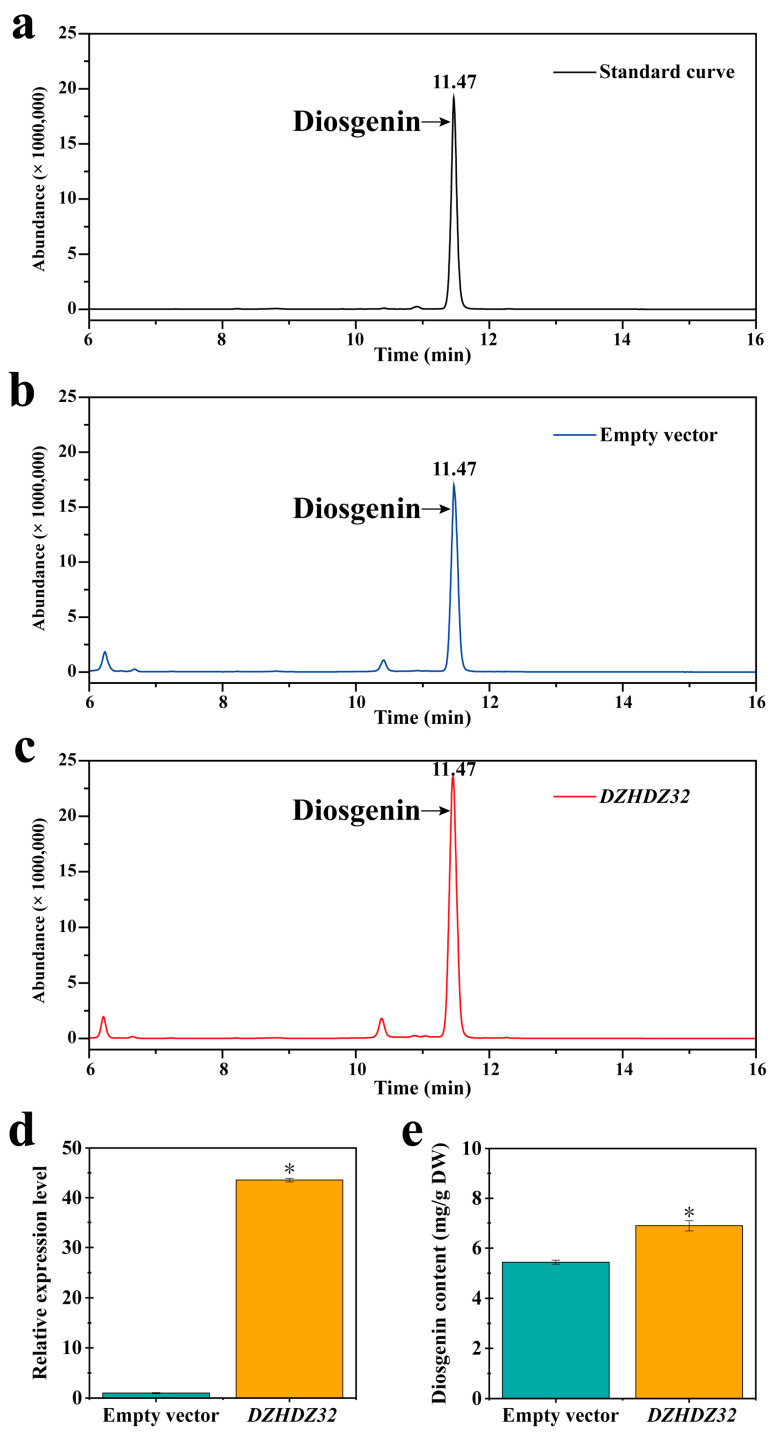
Transient overexpression of *DZHDZ32* promotes diosgenin accumulation and upregulates the expression of diosgenin biosynthesis pathway genes in the leaves of *D. zingiberensis*. (**a**) Standard curve of diosgenin (10 μg/mg). (**b**) Diosgenin content of the transient overexpression of the empty vector control and (**c**) of the transient overexpression of *DZHDZ32*. (**d**) Expression levels of *DZHDZ32* and (**e**) content of diosgenin in leaves of transiently overexpressing *DZHDZ32* relative to the empty vector control. Three biological replicates were performed in this study. Student’s *t*-test: * *p* < 0.05.

**Figure 3 ijms-26-04185-f003:**
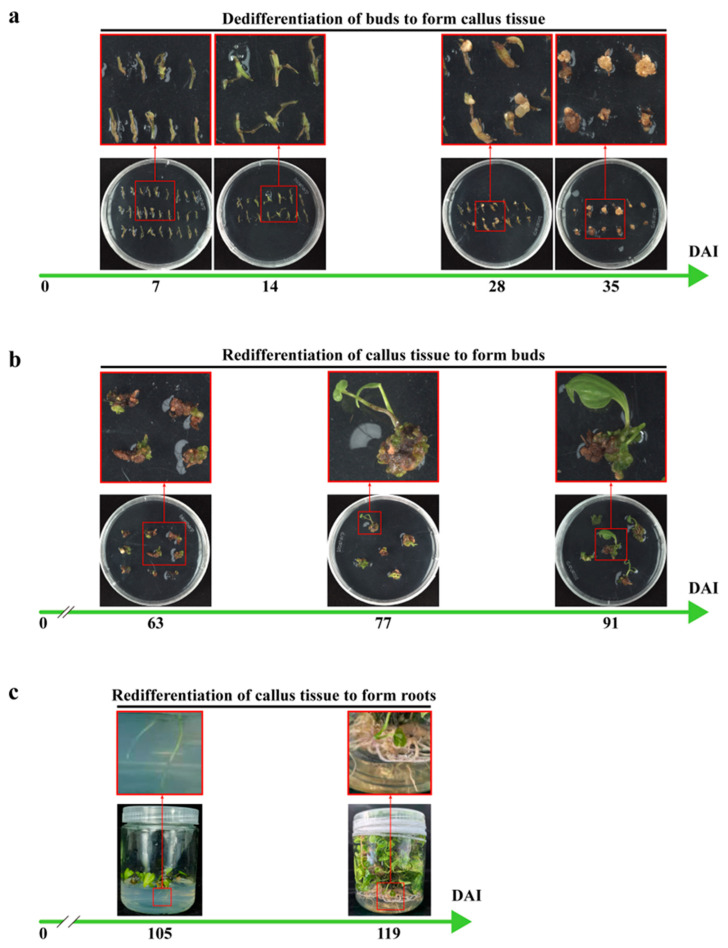
The time course of genetic transformation in *D. zingiberensis*. (**a**) After *Agrobacterium* infection of buds, explants were transferred to a co-cultivation medium, then to a callus induction and selection medium, and subsequently to a callus subculture and selection medium. The dedifferentiation and callus formation status at 7, 14, 28, and 35 DAI are shown. (**b**) Calli were moved to a regeneration and selection medium for shoot induction. Images show regenerated shoots at 63, 77, and 91 DAI. (**c**) Regenerated shoots were transferred to a rooting medium for root development, with images taken at 105 and 119 DAI.

**Figure 4 ijms-26-04185-f004:**
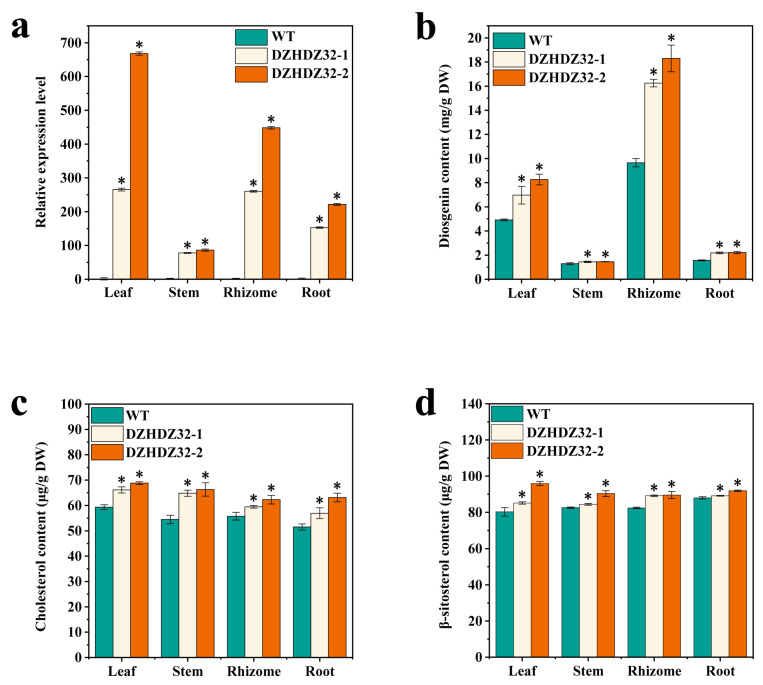
Levels of *DZHDZ32* expression, diosgenin, and its biosynthetic intermediates were analyzed in OE lines compared to WT plants. (**a**) Relative *DZHDZ32* expression was confirmed to be significantly higher in the roots, stems, leaves, and rhizomes of the OE lines than in WT, using *DZACTIN* as a control housekeeping gene. (**b**) Overexpression of *DZHDZ32* resulted in increased accumulation of diosgenin across all tested tissues: roots, stems, leaves, and rhizomes. Levels of the intermediate metabolites (**c**) cholesterol and (**d**) β-sitosterol were measured and compared in the leaves, stems, roots, and rhizomes of OE lines and WT. Data represent means ± SD (standard deviation) from three biological replicates. Student’s *t*-test: * *p* < 0.05.

**Figure 5 ijms-26-04185-f005:**
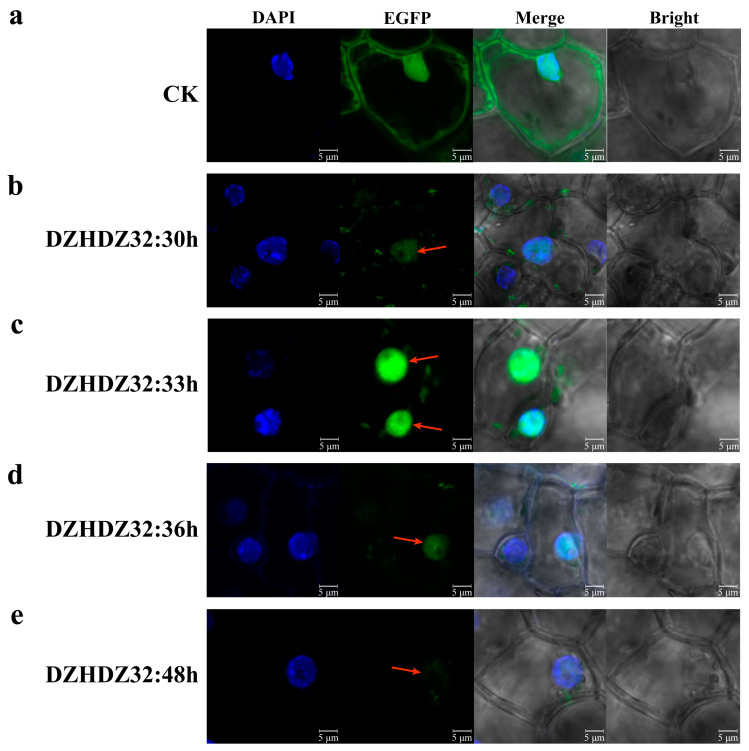
Subcellular localization of the DZHDZ32 protein in mesophyll cells of *D. zingiberensis*. (**a**) Confocal microscopy images of leaf mesophyll cells at 48 hpi with *Agrobacterium* carrying the empty vector PQG110 (35S: EGFP); this sample serves as the negative control. (**b**) Leaves expressing the DZHDZ32-EGFP fusion protein are shown at 30 hpi, (**c**) at 33 hpi, (**d**) at 36 hpi, and (**e**) at 48 hpi. The red arrow indicates the fluorescence signal of DZHDZ32-EGFP. Three biological replicates were performed in this study. DAPI staining was used to mark nuclear localization. Bar = 5 μm.

**Figure 6 ijms-26-04185-f006:**
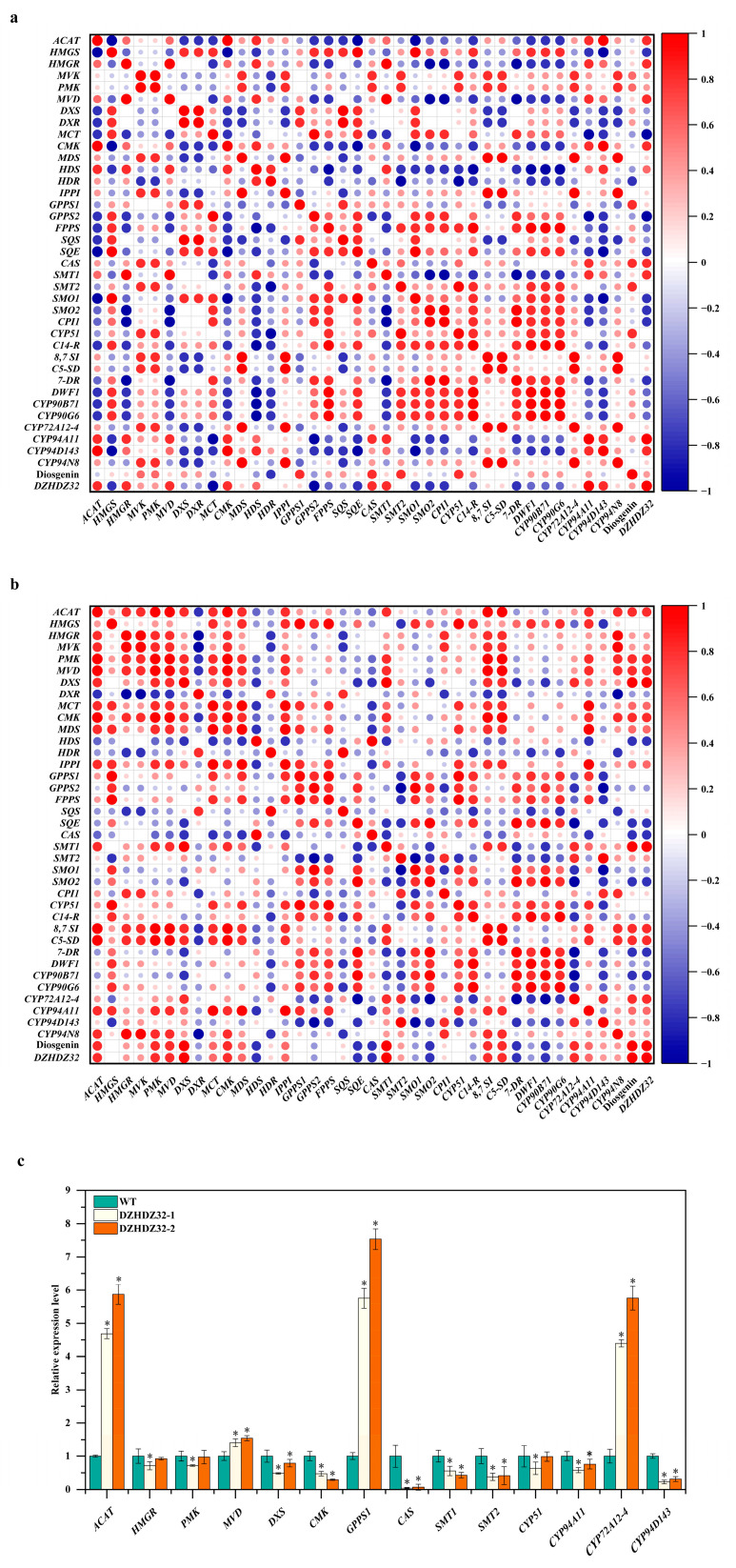
Correlation analysis of the expression of *DZHDZ32* with genes in the diosgenin biosynthetic pathway under drought and MeJA treatments and relative expression levels of genes involved in diosgenin biosynthesis. (**a**) Correlation analysis of the expression of *DZHDZ32* with genes involved in the diosgenin biosynthetic pathway in *D. zingiberensis* treated with drought and (**b**) with MeJA. Red and blue dots indicate positive and negative correlations, respectively; dot size indicates the correlation coefficient value. (**c**) Relative expression levels of genes involved in diosgenin biosynthesis in the *OE* lines and WT. Data represent the mean ± SD of three biological replicates. Student’s *t*-test: * *p* < 0.05.

**Figure 7 ijms-26-04185-f007:**
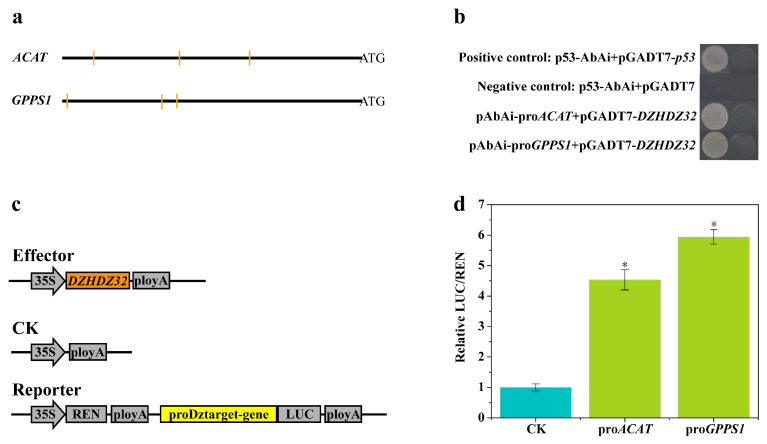
DZHDZ32 binds to the promoter regions of several diosgenin biosynthesis pathway genes and regulates their expression. (**a**) Binding site analysis of the upstream 2000 bp of diosgenin biosynthesis pathway genes related to DZHDZ32. The black horizontal lines represent the 2000 bp upstream sequence of pathway genes, and the orange vertical lines represent the predicted DZHDZ32 binding sites. (**b**) Y1H (yeast one-hybrid) analysis of DZHDZ32 binding to the promoters of diosgenin biosynthesis pathway genes related to DZHDZ32. (**c**) Construction of vectors in the dual-luciferase assay. DZHDZ32, represented by an orange rectangle, was subcloned into the pGreenII62-SK vector (effector), and CK was the pGreenII62-SK empty vector. The promoter regions of some diosgenin biosynthesis pathway genes represented by a yellow rectangle were subcloned into the pGreenII0800-LUC vector (reporter) upstream of the luciferase reporter gene. (**d**) Dual-LUC (dual-luciferase assay) analysis of DZHDZ32 binding to the promoters of diosgenin biosynthesis pathway genes. Expression levels were determined by the LUC/REN ratio. Three biological replicates were performed in this study. Error bars represent standard error. Student’s *t*-test: * *p* < 0.05.

**Figure 8 ijms-26-04185-f008:**
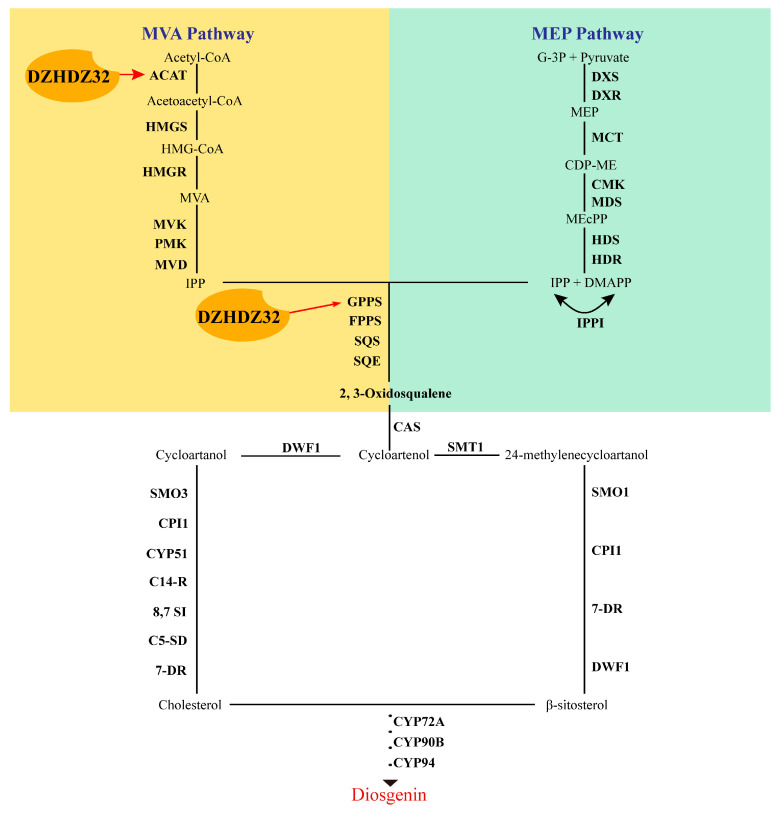
Model diagram of DZHDZ32 regulating diosgenin biosynthesis genes. Red arrows indicate the target genes of transcription factor DZHDZ32.

**Table 1 ijms-26-04185-t001:** Comparison of the efficiency of existing genetic transformation protocols in *Dioscorea* species.

Plant Species	Transformation Efficiency	Time (From *Agro*-Infection to Regeneration of Complete Transgenic Plant)	Target Gene	Reference
*D. zingiberensis*	46.8%	4 months	*DZHDZ32*	In this study
*Dioscorea rotundata*	9.4 to 18.2%	3–4 months	*GUSA*	[24]
*D. zingiberensis*	Very low	Unknown	*DZFPS*	[22]
*Dioscorea alata* L.	2.33%	Unknown	*DRPDS*	[23]

## Data Availability

All data are presented inside the manuscript and its Appendix A.

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
