# Peer review of "An HD-ZIP I Transcription Factor DZHDZ32 Upregulates Diosgenin Biosynthesis in Dioscorea zingiberensis"

_ijms, 2025, doi:10.3390/ijms26094185_

Round 1

Reviewer 1 Report

Comments and Suggestions for Authors

In this article, the authors identify that the overexpression of the hdzip gene called DzHDZ3, via an optimized genetic transformation protocol, upregulates diosgenin biosynthesis in Dioscorea zingiberensis.

I have several issues

Abstract: I suggest providing more detail about the relevance of Dioscorea zingiberensis and diosgenin. It would be beneficial to explain why this plant and molecule are important compared to other models.

Lines 41, 44, 49:Please define the gene abbreviations. Proteins should be written in uppercase, while genes should be in italics and uppercase. This should be checked throughout the text.

Introduction:Please provide a more thorough description of diosgenin, including its type of molecule and significance in biochemical synthesis. I would also like to see more background on how HDZIP genes regulate the biosynthesis of compounds related to diosgenin. It is not suitable to cite “speculative studies” that propose the regulation of diosgenin by HDZIP genes.

Lines 90-95: This information should be included in the introduction.

Line 97:Please specify the meaning of the gene DzHDZ3.

Figure 1a: Please indicate the number of amino acids in DzHDZ3 and in its domains.

Figure 2: I suggest moving the line that indicates the empty vector and the overexpression line, as it seems to indicate a value on the y-axis. The value for the overexpression line is the same as that of the empty vector.

Lines 123 and Figure 3: Please provide a more detailed explanation regarding the modifications and improvements made to the genetic transformation system of D. zingiberensis.

Lines 138-139: I recommend placing this information at the beginning of the section to highlight the study's relevance.

Table 1: Please provide a more descriptive title related to the comparison of different studies on transformation efficiency.

Figure 3: The image appears incomplete; please enhance the figure caption for clarity.

Figure 4: Some words are duplicated. Please indicate the control housekeeping genes used for the expression analysis.

Figure 5: Clearly indicate the negative control in the image. The figure caption needs improvement; consider reordering the description to facilitate analysis and eliminate duplicate words.

Line 235: I do not understand the title “Correlation analysis of DzHDZ3 with genes involved in the diosgenin biosynthetic pathway.” I suggest adding the word “expression” to ensure a better understanding that this analysis is related to the expression association of different genes. The same clarification s

hould be included in Figure 6.

Comments on the Quality of English Language

Please check throughout the text that there are no duplicated words.

In some cases, an improvement of the ideas for a better understanding is necessary.

Reviewer 2 Report

Comments and Suggestions for Authors

Authors have identified DzHDZ3 as a potential regulator of diosgenin biosynthesis in Dioscorea zingiberensis through transient overexpression. Authors have also optimized genetic transformation method for D. zingiberensis and generated two DzHDZ3-overexpressing lines. Diosgenin content is significantly increased in leaves compared to wild-type plants. Yeast one-hybrid and dual luciferase assays demonstrated the interaction of the genes in the overexpressed lines and identified DzHDZ3 as a key regulator of diosgenin biosynthesis. This study is important and has significant impact on drug development. Authors are advised to address the following comments that would improve the manuscript,

  1. Figure 2. d,e: Correct the type in the Empty vector title in the x-axis.
  2. Table 1. What is the major changes in the current study that contributed for the increase in the transformation efficiency to 46% compared to the other previous studies in Dioscorea.
  3. The legend for Figure3 suggested to enhance with more details.
  4. What are the differences between DzHDZ3-1 and DzHDZ3-2 with respect to the gene size, number of exons or any other differences. Figure4: I see significant differences int eh expression level but not in the diosgenin levels.
  5. Line 97: How it was identified transcription factor gene DzHDZ3 that can promote diosgenin accumulation?

Overall, the study is conducted comprehensively and significant to decode the role of DzHDZ3 in diosgenin accumulation

Round 2

Reviewer 1 Report

Comments and Suggestions for Authors

I have observed that the authors have put significant effort into improving the manuscript and addressing my suggestions. My final recommendations include using italics and uppercase letters for gene names, and uppercase for protein names. Additionally, there are inconsistencies in formatting; for instance, AP2/ERF (APETALA2/ethylene responsive factor) includes the full name in parentheses, while CONSTITUTIVE PHOTOMORPHOGENIC DWARF (CPD) does not. Please review the text for consistency.

 I have also noticed some incomplete words, such as “re-sponses” in line 63. Furthermore, figure 7 appears distorted such as a mirror image, making the names difficult to read

.

Comments on the Quality of English Language

Not big issues detected

.

Author Response

Comments 1: My final recommendations include using italics and uppercase letters for gene names, and uppercase for protein names.

Response 1: Thank you for your comments. We have italicised the gene names (line 109, 113, 116, 226, 240, 506, and 507) and kept the protein names in standard font (line 105,

305, 307, 308, 312, 334, 338, 342, 344, 353, 365, 381, 383, 384, 387, 561, 562, and 564).

All gene names and protein names have been formatted in uppercase letters. (line 2, 12, 15, 17, 22, 26, 27, 29, 50, 66, 103, 105, 109, 113, 116, 119, 120, 122, 123, 127, 130, 131, 132, 147, 192, 197, 198, 200, 201, 203, 211, 212, 214, 219, 221, 223, 226, 231, 235, 236, 240, 244, 245, 249, 252, 255, 258, 261, 264, 265, 268, 277, 281, 284, 286, 292, 293, 300, 301, 304, 305, 307, 308, 312, 315, 317, 319, 323, 325, 327, 328, 329, 334, 337, 338, 340, 342, 344, 348, 353, 355, 365, 367, 375, 381, 383, 384, 387, 392, 400, 401, 403, 418, 419, 420, 453, 505, 506, 507, 516, 523, 529, 538, 540, 558, 559, 561, 562, and 564)

Comments 2: There are inconsistencies in formatting; for instance, AP2/ERF (APETALA2/ethylene responsive factor) includes the full name in parentheses, while CONSTITUTIVE PHOTOMORPHOGENIC DWARF (CPD) does not.

Response 2: Thank you for your comments. We have revised the formatting to include the full names within parentheses, as suggested. (line 50, 51, 52, 68, 69, 70, 111, 168, 172, 173, 241, 242, 246, 247, 359, 360, 509, 510, 550 and 551)

Comments 3: I have also noticed some incomplete words, such as “re-sponses” in line 63.

Response 3: Thank you for your comments. We corrected the incorrectly hyphenated word “re-sponses” to “responses”. (line 63)

Comments 4: Furthermore, figure 7 appears distorted such as a mirror image, making the names difficult to read.

Response 4: Thank you for your comments. We have correctly inserted Figure 7 into the manuscript in the appropriate format. (line 322)

4. Response to Comments on the Quality of English Language

Point 1:

Response 1: The positions for language improvement in the article have been highlighted in red.